# A Comparison Study of Data Link with Medium-Wavelength Infrared Pulsed and CW Quantum Cascade Lasers

Janusz Mikołajczyk 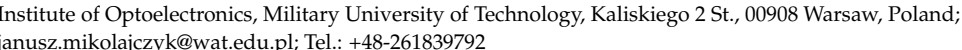

Institute of Optoelectronics, Military University of Technology, Kaliskiego 2 St., 00908 Warsaw, Poland; janusz.mikolajczyk@wat.edu.pl; Tel.: +48-261839792

**Abstract:** In this paper, a comparison study of a quantum cascade laser used for signal transmission by free-space optics is presented. The main goal is to define the capabilities of medium-wavelength infrared lasers operated in pulsed or continuous wave (cw) mode through testing and analyzing a laboratory setup of a data link operated at wavelengths of 4.5 µm (pulsed, peak power 3 W) and 4.8 µm (cw, average power ~20 mW). In this spectral range, the link budget is also defined by radiation attenuation in the atmosphere (absorption, scattering, and turbulence interaction). The performed measurements define unique operational aspects of the quantum cascade lasers considering on–off keying modulation. The registered light pulse changes for different parameters of driving current signals determine some limitations in both rate and data range. Finally, we present eye diagrams of the signals obtained using two data links.

**Keywords:** free-space optics; quantum cascade lasers; wireless communications; quantum cascade lasers

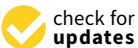



## 1. Introduction

Free-space optics (FSO), also referred to as optical wireless communication (OWC) with Gbit data rate, is an attractive alternative to radio technology. The first device constructed with FSO was the photophone constructed by Alexander Graham Bell, in 1880, to transmit sound over a distance of 213 m [1]. More recently, there has been growing interest in this technology. Compared to other wireless technologies, it provides a high data rate, free-charge spectrum, low cost, simple deployment, low power consumption, no-interference of electromagnetic (EM) signals, and a high security level [2]. However, its high susceptibility to signal attenuation in the atmosphere is a significant link range limiting factor [3] that is defined by the weather, turbulence conditions, and other parameters. The light propagation losses are described by absorption, scattering, or dispersion phenomena and the sources of these factors are, for example, atmospheric compounds, mist, fog, rain, snow, haze, dust, or air turbulence [4]. Nowadays, FSO systems use near infrared (NIR, 0.7–1.4 µm) or short wavelength infrared (SWIR, 1.4–3 µm) radiation to transmit information as the "last mile" link [5]. These devices enable transmission of tens of Gbps (gigabits per second) over several kilometers, where the installation of other communication technologies (fiber) is too expensive or practically impossible [6]. In today's market, FSO systems are available with a data rate up to 30 Gbps at a distance of ~1.5 km [7]. However, currently, the practical challenge for developing devices with FSO is to provide 10 Gbps at a distance over 5 km [6]. Some experimental setups are also being tested to achieve a data range of terabits per second at a distance of several meters, and over several kilometers in outer space [8].

The main limitation of technological applications of FSO is still the atmospheric attenuation of optical radiation. The performed analyses indicate that longer-range FSO may become feasible using longer wavelength ranges (medium-wavelength infrared (MWIR) of 3–5 µm or long-wavelength infrared (LWIR) of 8–12 µm) to significantly increase the transmit radiation power without surpassing regulatory safety limits ("eye-safe"). Working

at MWIR or LWIR wavelengths also minimizes the influence of photon scattering effects, defined as the relation between the size of the scattering particle and the radiation wavelength. For many years, there were no radiation sources that operated in these ranges characterized by compact construction, high modulation rate, room temperature operation, and low power consumption. In particular, the design of optoelectronic cascade structures (quantum cascade (QC) lasers and inter-band cascade (IC) lasers) has defined a new direction in the development of FSO systems that operate in a MWIR spectral range. Nowadays, the performance of state-of-the-art systems indicate that the technological readiness of these systems is related to laboratory devices. In Table 1, some FSO systems are listed.

**Table 1.** Free-space optics (FSO) systems operated in medium-wavelength infrared (MWIR) range.

| Wavelength | Laser | Data Rate/ Modulation Rate | Modulation Type * | Distance | Temp. | Year |
|---|---|---|---|---|---|---|
| 3.0 μm | ICL | 70 Mb/s | NRZ-OOK | 1 m | 77 K | 2009 [9] |
| 4.7 μm | QCL | 40 MHz | Analog | 2.5 m | 293 K | 2015 [10] |
| 4.65 μm | QCL | 3 Gb/s | NRZ-OOK, PAM-4, PAM-8 | 5 cm | 293 K | 2017 [11] |
| 4.65 μm | QCL | 4 Gb/s | PAM-4, DMT | 5 cm | 293 K | 2017 [11] |
| 4.8 μm | QCL | 10 MHz | Analog | a few m | 293 K | 2020 [12] |
| 3.0 μm | ICL | 2 GHz | Analog | x cm | 77 K | 2009 [13] |
| 3.0 μm | ICL | 70 Mb/s | OKK | x cm | 293 K | 2010 [14] |
| 3.6 μm | DFG | 200 Mb/s | OKK | 10 m | 293 K | 2017 [15] |

* NRZ-OOK, on–off keying with non-return-to-zero coding format; PAM-x, x-level pulse amplitude modulation; DMT, discrete multitone modulation; DFG, difference frequency generation laser.

QC lasers are applied in most FSO systems. The aim of their construction is to achieve the highest data rates, which are usually obtained using cw lasers driven by DC biasing current and radio frequency (RF) modulation signal (T-bias electronic circuits). In this configuration, directed modulation at a few GHz can be obtained. The faster modulation speeds require specially designed devices, injection of RF signal, or cryogenic cooling.

The main goal of this study is to define the capabilities of two MWIR QC lasers designed for pulsed or cw mode operation. According to their radiation wavelength (~4.5 μm), comparable link budgets were performed considering different data sources of radiation attenuation, for example, the Kim model with visibility, the HITRAN molecular absorption database, and the PcModWin 6.0 software (Ontar Corporation, North Andover, USA) with environmental models. For a comparison, the same budget was calculated for the link operated at a wavelength of 1.5 μm which corresponds to commercially available FSO systems.

Finally, tests of newly designed FSO systems using a "shelf-ready" pulsed QC laser (~3 W peak power and ~4.5 μm wavelength) and a cw QC laser (~20 mW average power and ~4.8 μm wavelength) were conducted during operation at room temperature. The "eye diagram" analyses indicated that both QC lasers can be used in an FSO system providing a direct current modulation with a rate of 6 MHz (for pulsed laser) and 100 MHz (for cw laser). Additionally, the range calculations for these FSO systems indicated the need for a precise analysis of the spectral characteristics of both radiation source and atmospheric attenuation. It can significantly modify the data link range. For the designed links, which are spectrally separated by 300 nm, the high difference in QC lasers power has not so significant influence on the link distance.

During the test of the cw laser, a significant relationship between amplitude and the frequency of light pulses was also identified. In practice, considering the speed and range of FSO links, a compromise should be established. In the case of the tested high-power pulsed laser, the direct modulation rate is the highest one using such lasing structures. For both lasers, the shapes of the switching current include some disturbances (oscillations) caused by the electrical parameters of their signal interfaces. Therefore, a dedicated technology of these interfaces or switching electronics integration is critical for high-speed QC laser applications.

Summarizing, the paper describes a practical study of a state-of-the-art MWIR quantum cascade laser technology considering the application of both cw and pulsed lasers in free space data transmission.

## 2. Materials and Methods

Optical radiation is a physical signal that transmits data in free space. Light is modulated and emitted by a transmitter, propagated through the air channel, and registered by a receiver (Figure 1).

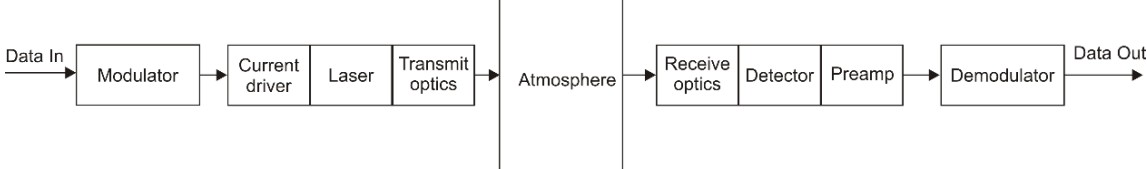

**Figure 1.** Block scheme of FSO operation.

The power of the detected signal, $P_d$, is determined by features of these elements usually described as follows:

$$P_d = P_0 \frac{D_d^2}{(D_0 + \theta_{div}L)^2} e^{-\gamma(\lambda)L},$$

(1)

where $P_0$ is the laser light power, $\gamma(\lambda)$ is the extinction coefficient, $L$ is the link distance, $\theta_{div}$ is the beam divergence, and $D_0$ and $D_d$ are the optics diameters of the transmitter and receiver, respectively. To make a connection, there is no need to define parameters of the transmission channel, but the line-of-sight positioning is required for transceiver units.

The beam attenuation described by the extinction coefficient is affected by absorption and scattering phenomena resulting from light interaction with air molecules and aerosol particles. Light sources working in the spectral range called "atmospheric windows" minimize the influence of radiation absorption. The atmospheric transmission at a distance of 1 km obtained using the HITRAN database is presented in Figure 2.

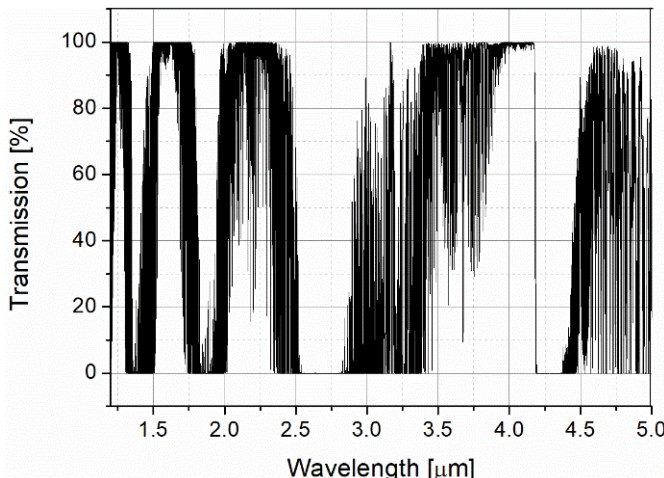

**Figure 2.** Atmospheric transmission at a distance of 1 km (from the HITRAN database).

In a high transparent spectral range, the operation of an FSO system is limited by scattering (Rayleigh, Mie, and geometrical). To analyze this phenomenon, scattering particle size should be compared to the light wavelength (Table 2).

**Table 2.** Typical atmospheric scattering phenomena.

| Type | Air Molecules | Aerosol | Fog | Raindrops | Snow |
|---|---|---|---|---|---|
| Radius (μm) | $10^{-4}$ | $10^{-1}$ to 1 | 1 to 10 | $10^2$ to $10^4$ | $10^3$ to $5 \times 10^3$ |
| Concentration (cm$^{-3}$) | $<3 \times 10^{19}$ | $10^{-1}$ to $10^4$ | 10 to 100 | 10 to $10^3$ m$^{-3}$ | 0.01–2 g/m$^3$ |
| Scattering regime | Rayleigh | Rayleigh, Mie | Mie, geometrical | Geometrical | Geometrical |

For MWIR radiation, the scattering effects are mainly described using Mie theory and metric based on "visibility (Vis)" [16]. An international visibility code has also been defined [17] with some weather effects (Table 3).

**Table 3.** International visibility code.

| Weather Conditions | Dense Fog | Thick Fog | Moderate Fog | Very Light Fog | Light Haze | Very Light Haze | Clear Air |
|---|---|---|---|---|---|---|---|
| Visibility (Vis) (m) | 50 | 200 | 500 | 1000 | 4000 | 10,000 | 23,000 |
| Attenuation (dB/km) | 315 | 75 | 28.9 | 13.8 | 3.1 | 1.1 | 0.47 |

In practice, visibility is usually read from weather data. Visibility and wavelength can both be used to estimate the scattering coefficient β(λ). It was described empirically by Kruse [18] as follows:

$$\beta(\lambda) = \frac{3.91}{\text{Vis}} \left( \frac{\lambda}{\lambda_0} \right)^{-q}, \tag{2}$$

where $\lambda_0$ = 550 nm is the reference wavelength, and $q$ is the particle size distribution coefficient [19]. In Figure 3, the calculated scattering coefficient for two wavelengths and different levels of visibility is determined (Figure 3).

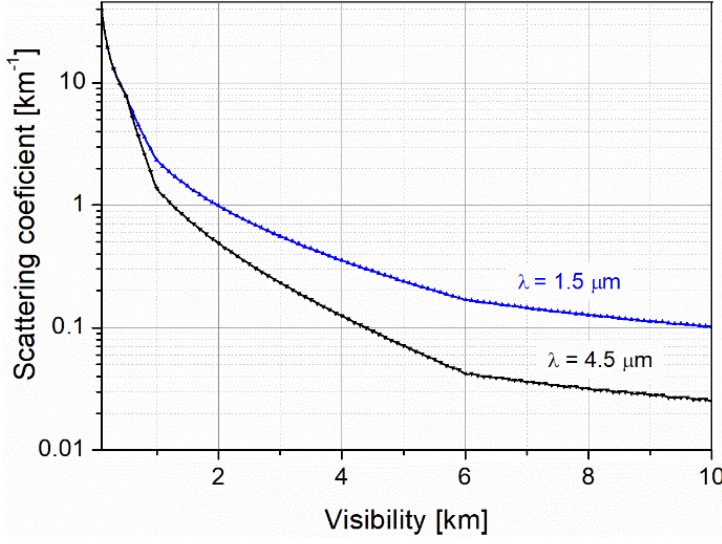

**Figure 3.** Scattering of light as a function of different visibilities for wavelengths 1.5 and 4.5 μm.

Commercially available FSO systems use a radiation of 1.5 μm wavelength; however, longer wavelengths are preferable in the case of light scattering. The mentioned descriptions of scattering are only some approximations of engineering calculations. For example, calculations of fog attenuation are more complex and distinguish the type of impact, i.e., radiation or advection. For advection fog, wavelength dependence of attenuation is linear. The quadratic spectral coefficient is a characteristic of radiation fog and higher light losses are observed. Additionally, the formation of fog is often accompanied by an increase in humidity [20]. The HITRAN database analysis shows that for a wavelength of 4.5 μm, the air transmission at a distance of 1 km is at least 22% lower as compared with a wave-

length of 1.5 μm. For rain and snow, the relationships between particle sizes and radiation wavelengths indicate no spectral benefits [21]. The analysis of the influence of dust shows that signal attenuation is strong because its particle sizes are comparable with the FSO radiation wavelength. This attenuation is even seven times higher than that of fog. With visibility changes from 1 to 0.2 km, attenuation is increased from 50 to 300 dB/km [22]. To summarize the analyses of the radiation attenuation for different weather conditions, some simulations of light propagation at wavelengths of 1.55 and 4.5 μm were performed using the PcModWin 6.0 software. The weather conditions were defined by visibility and some kinds of aerosols. The simulation results are listed in Table 4.

**Table 4.** Extinction coefficient for different weather conditions using the PcModWin 6.0 software [23].

| | | Aerosol, Vis | | | | | | | | |
|---|---|---|---|---|---|---|---|---|---|---|
| | | Only Absorption | Country, 23 km | Country, 5 km | City, 5 km | Maritime, 23 km | Desert, 23 km | Radiant Fog 0.5 km | Advection Fog, 0.2 km | Haze, 2 mm/h | Medium Rain, 12.5 mm/h |
| $\gamma$ (1/km) | 1.5 μm | 0.20 | 0.24 | 0.40 | 0.43 | 0.31 | 0.22 | 8.92 | 20.29 | 0.86 | 2.08 |
| | 4.5 μm | 0.24 | 0.25 | 0.31 | 0.33 | 0.30 | 0.26 | 9.24 | 21.58 | 0.87 | 2.10 |

For clear air and high visibility, light attenuation was determined by absorption and the level was comparable for the two wavelengths. In rain and very low visibility (below 500 m), the same beam attenuation was also noted. However, if visibility was increased, the light conditions of propagation at a wavelength of 4.5 μm were better.

Air turbulence can generate random variations in the refractive index modifying optical path performances [24]. As a result, the beam profile is redistributed, corresponding to beam wander, beam spreading, or scintillation.

The signal-to-noise ratio (SNR) can also be randomly changed by scintillations. It is characterized by the scintillation index corresponding to the normalized fluctuation of the signal as follows:

$$\sigma_I^2 = \frac{\langle I^2 \rangle - \langle I \rangle^2}{\langle I \rangle^2}, \qquad (3)$$

where I is the light intensity on the detector surface, and <I> is its mean value. For the condition of weak turbulence, this signal variance is determined by the refractive index structure parameter ($C_n^2$ index), link range (L), and light wavelength (k = 2π/λ) as follows:

$$\sigma_I^2 = 1.23 C_n^2 k^{\frac{7}{6}} L^{\frac{11}{6}}. \qquad (4)$$

The $C_n^2$ parameter defines the level of turbulence. Operation in longer wavelengths leads to better transmission during turbulence [25]. However, strong turbulence causes the same attenuation for both FSO wavelengths. Practically, the effect of scintillation can be minimized but not eliminated. The nature and level of scintillations depend on many factors, for example, temperature and pressure distribution, humidity, altitude, and surface type. All parameters and their impact are difficult to predict. However, some mathematical models of the effects have been developed. The influence of scintillation on the FSO data range is calculated according to the analytical dependences described in The Infrared Handbook [26] (by W.L. Wolfe and G.J. Zissis) and the Field Guide to Atmospheric Optics (by L.C. Andrews) [27]. In Figure 4, the FSO data link ranges providing SNR = 10 for different levels of turbulence are shown.

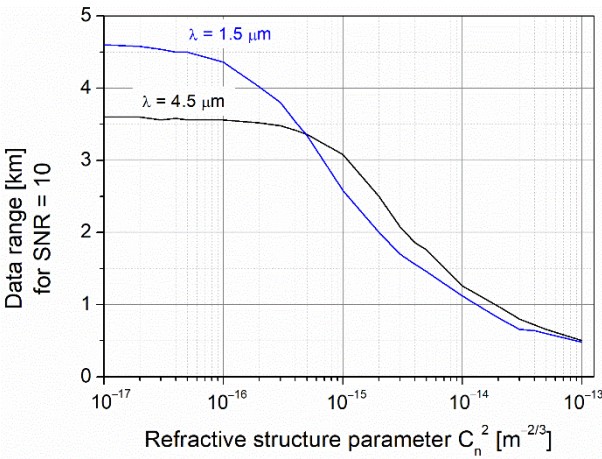

**Figure 4.** FSO link range vs. scintillation level for two wavelengths.

For weak and very weak turbulence, the data range is constant (above 3.5 and 4.5 km) for 4.5 and 1.5 μm wavelengths. If the refractive structure parameter increases above $5 \times 10^{-16}$ m$^{-2/3}$, the range starts decreasing, but the longer radiation wavelength provides an extended range.

To analyze the operation of an OWC transmitter with QC lasers in MWIR spectral range, two different quantum cascade structures were designed at the Institute of Electron Technology (Warsaw, Poland). The Laser #356 was a high-power pulsed QC laser (peak power 3 W and max. duty cycle of 8%) with a wavelength of ~4.5 μm. The second lasing structure, the Laser #253 was designed to operate in cw mode. It was characterized by a mean average power of 20 mW with a wavelength of ~4.8 μm (Figure 5a). The optical power-current (L-I) and spectral characteristics of these lasers are presented in Figure 5b.

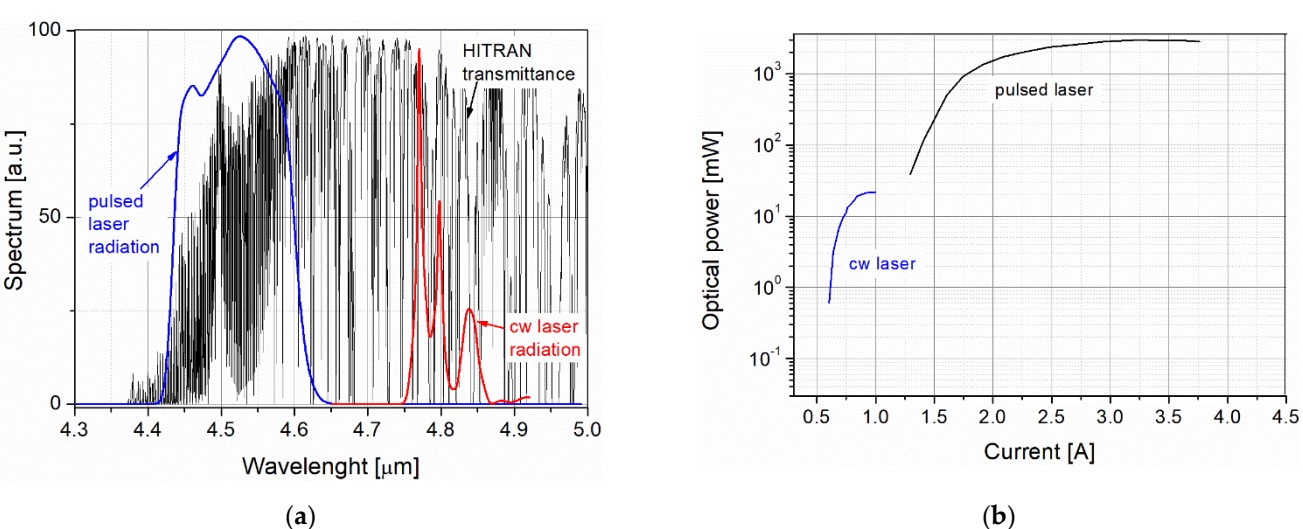

(a)                                    (b)

**Figure 5.** (**a**) Spectrum and (**b**) optical power-current (L-I) characteristics of both quantum cascade (QC) lasers.

For laser radiation, the same current driver was applied. The light beam was formed using an off-axis parabolic mirror with a three-inch diameter and a two-inch focal length. This setup collimated ~75% of the laser output power. The collimated beam was collected on the mercury-cadmium-telluride (MCT) photodetector with a four-inch parabolic mirror. The photodetector was mounted with an amplifier in a detection module fabricated by the VIGO System company. To determine the data link range, it was necessary to define several parameters of the FSO components (Table 5).

**Table 5.** Parameters of the FSO components.

| Parameter | Value |
|---|---|
| cw Laser power | 20 mW |
| Pulsed laser power | 3 W |
| Beam divergence | 1 mrad |
| Diameter of receiving optics | 100 mm |
| Detection module D* | $3 \times 10$ cmHz$^{1/2}$/W |
| Detector surface | $0.5 \times 0.5$ mm$^2$ |
| Bandwidth | 700 MHz |

D* is the symbol of normalized detectivity

To determine radiation attenuation of these two lasers, considering the characteristics of atmosphere absorption spectra with many absorption lines, more precise analyses of light attenuation was performed. The analyzed QC lasers are characterized by broad lines of emitted radiation (~100 nm) in the spectra and the maximum changes with supplying current. The determination of their light transmission in the atmosphere should be based on the average values of the extinction coefficient in these spectra. However, the results cannot be directly referenced as maximum extinction coefficient because of the strong influence of narrow $H_2O$ absorption lines in this spectral ranges. The averaged wavelength ranges were 4.45–4.55 µm and 4.75–4.85 µm, for the pulsed laser and the cw laser, respectively. It was assumed that there were no differences in the scattering effect for these radiations because of their comparable wavelengths. The averaged extinction coefficient values for summer and winter at the latitude corresponding to Poland are presented in Figure 6. The environmental parameters were temperatures of 20.5 and $-1.5$ °C and water vapors of 13.4 and 3.4 g/m$^3$.

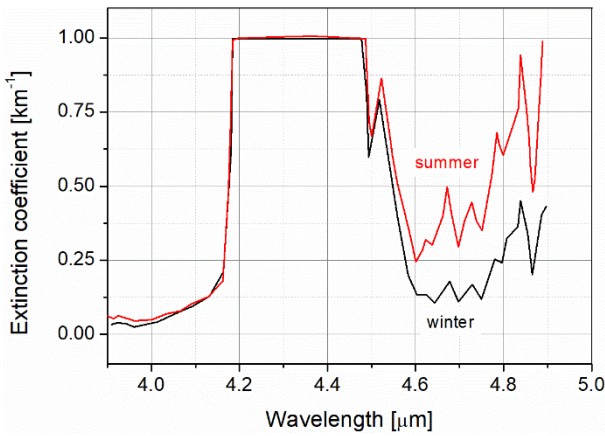

**Figure 6.** Averaged extinction coefficient for summer and winter.

According to these characteristics, the values of extinction coefficient used in the link budgets are listed in Table 6.

**Table 6.** Extinction coefficients for two lasers simulated for two weather seasons.

| Laser | Summer | Winter |
|---|---|---|
| Laser #356 (4.5 µm) | 0.75 km$^{-1}$ | 0.70 km$^{-1}$ |
| Laser #253 (4.8 µm) | 0.65 km$^{-1}$ | 0.25 km$^{-1}$ |

The analyses of the data link range were performed using the previously mentioned analytical model, data of the FSO components, and atmospheric attenuation listed in Tables 5 and 6. In Figure 7, some obtained characteristics of SNR values are presented.

For no scintillation, better transmission ranges were obtained for the pulsed laser with 150-times higher peak power but they are not so significant (only twice).

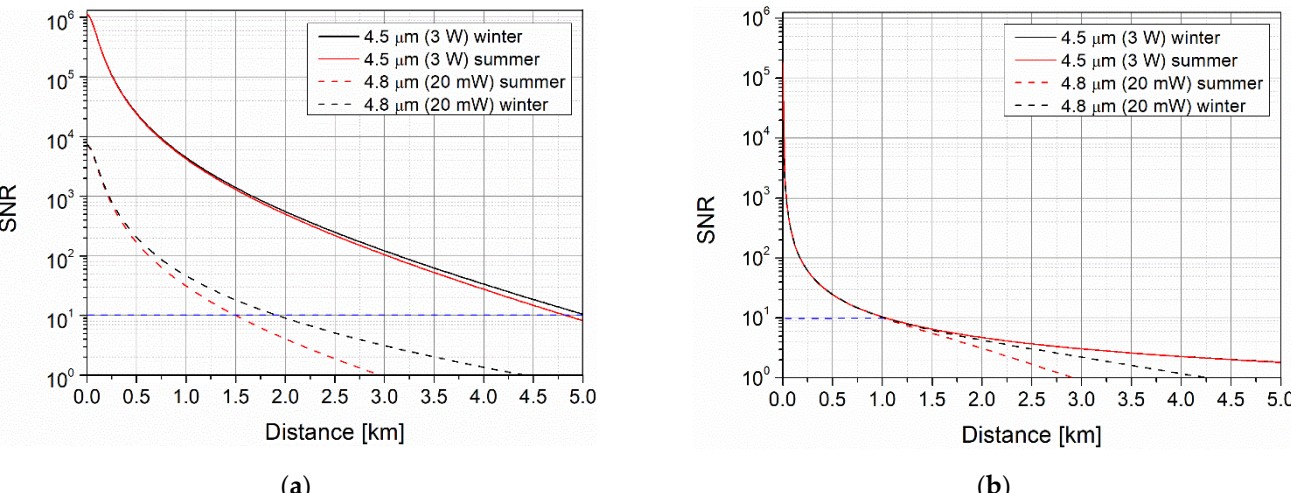

(**a**)                                                      (**b**)

**Figure 7.** The signal-to-noise ratio (SNR) value vs. data link range in the case of (**a**) no scintillation and (**b**) medium scintillation $C_n^2 = 10^{-14}$ m$^{-2/3}$.

The data link ranges were 1.5–5 km for the cw and pulsed lasers, respectively. For strong scintillation, the link distance decreased significantly below ~1.0 km for both lasers. An increase in radiation power did not improve the data link range. Although the average radiation power increased, its fluctuations also increased and, as a result, the SNR value decreased.

## 3. Results

During the tests of a free space data link, laser structures were placed in an LHH model housing device (Alpes Lasers S.A., Switzerland) which connected a driving current signal to a laser structure placed on a standard copper submount and controlled the temperature using a two-stage TEC module. The triggering signals for the current driver were generated in a pattern signal generator model Picosecond and collimated with an off-axis parabolic mirror. To monitor laser current pulses, the current probe model CT-1 Tektronix (Tektronix Inc., Beaverton, USA) was also applied. The registered pulses were visualized and processed in an oscilloscope model MSO 6 Tektronix with a built-in eye diagram toolbox. The schematic setup of the FSO system is presented in Figure 8.

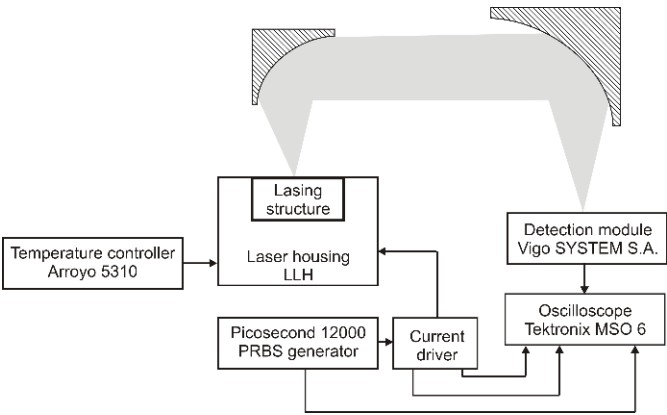

**Figure 8.** The schematic setup of the free space data link.

The main tasks of the preliminary tests were to define the limitations of the light pulses concerning pulse power, time duration, and frequency. An analysis was performed in room temperature conditions and the shapes of the driving current signal and the laser light power were compared. The shapes were a coincidence but some differences were observed in the rising pulse part. For both lasers, the light pulses are a bit faster and start later (delay time ~10 ns) (Figure 9), and therefore time was needed to exceed the lasing threshold current and to obtain population inversion. These time delays also adversely influenced the modulation rate. Some amplitude oscillations were also observed in both signals generated by impedance mismatching of the current driver and the laser interface.

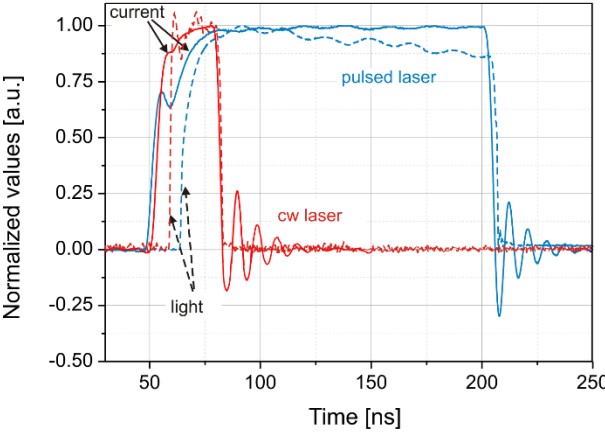

**Figure 9.** Laser light power and driving current pulses for pulsed and cw lasers.

The pulse characteristics of these lasers were determined for time durations of 18 ns (pulsed) and 30 ns (cw). Some registered pulses for different driving currents are presented in Figure 10. For the pulsed laser, it can be observed that there is a strong influence of current on amplitude and time duration of light pulses. By decreasing the current, a decrease in both optical pulse amplitude and its time duration (min. duration time ~5 ns) is noted. In the case of the cw laser, the driving current directly influences the radiation pulse power. However, the shape of these pulses (time duration) has not been changed. Although the average power of this laser is 20 mW, the obtained peak power for the duty cycle of 30% is ~220 mW. In this case, the pulsed operation mode of this laser results in a better data link.

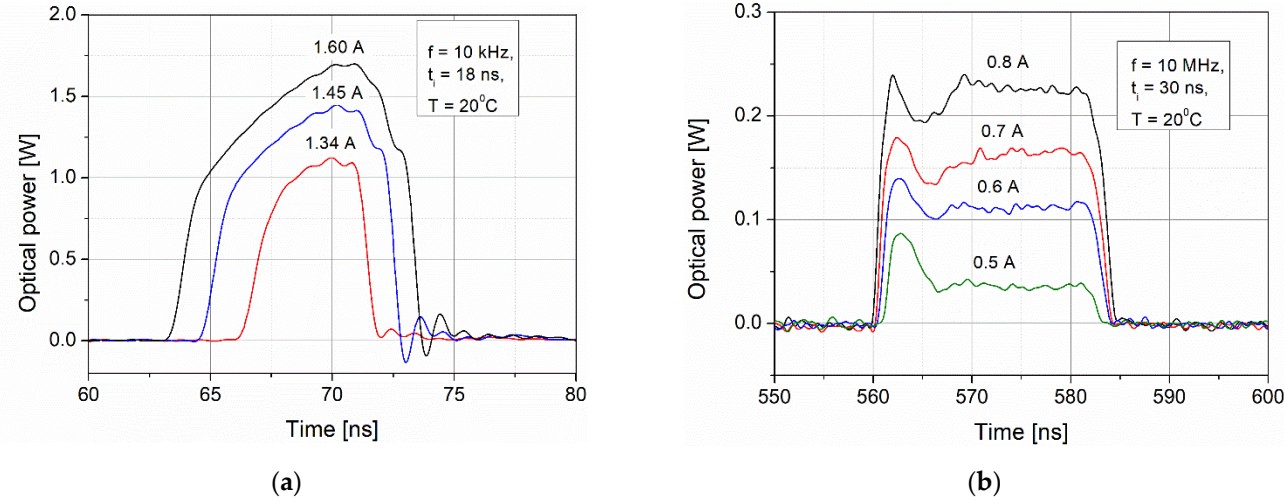

**Figure 10.** Light pulses for different driving currents for (**a**) pulsed and (**b**) cw lasers.

Finally, the influence of the pulse duty cycle on signal amplitude was also determined (Figure 11). The pulsed laser is more sensitive to duty cycle changes and a four-times higher duty cycle decreases the pulse power by about 30%. For the cw laser, the same decrease is measured for a 10-times higher duty cycle.

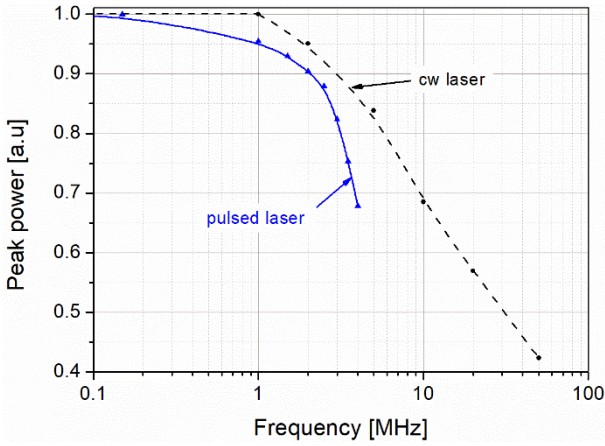

**Figure 11.** Peak power vs. pulse frequency (pulse duty cycle).

A preliminary test of information signal transfer was to send several pulses in a frame (BURST). The BURST consisted of 16 triggering pulses with a repetition rate up to a frequency of 2.5 MHz (pulsed) and 40 MHz (cw) (Figure 12). That frequency was limited by the efficiency of the cooling system. The measured triggering signal period was ~25 ns for the pulsed laser and ~10 ns for the cw laser. However, for the pulsed laser, the main limitation was the pulse duty cycle that caused the mentioned thermal heating. This decreased the amplitude of the next pulses in the frame signal. The amplitudes of the radiation pulses were constant with some oscillations in the case of the cw laser.

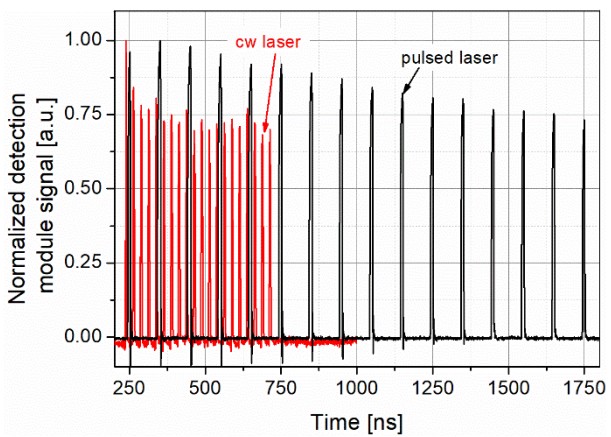

**Figure 12.** Registered pulses for BURST (16 triggering pulses with a repetition rate up to a frequency of 2.5 MHz (pulsed) and 40 MHz (cw)) mode operation of pulsed and cw lasers.

The final step of the QC laser testing was to analyze emissions by PRBS signals consisting of 10-bit frames ($1000010000_{BIN}$) at the distance of 2 m. That frame was generated in the generator model 12000 Picosecond and eye diagrams were visualized on the oscilloscope screen.

The registered diagrams for return-to-zero (RZ) coding format (pulsed) and non-return-to-zero (NRZ) coding format (cw laser) are presented in Figure 13, which correspond to modulation rates of 6 and 100 MHz, respectively. Although the cw laser could be modulated at a higher speed, the signal level was much lower than for the pulsed laser.

This describes a compromise between high transmit rate (defined by modulation rate) and long data link (defined by optical power).

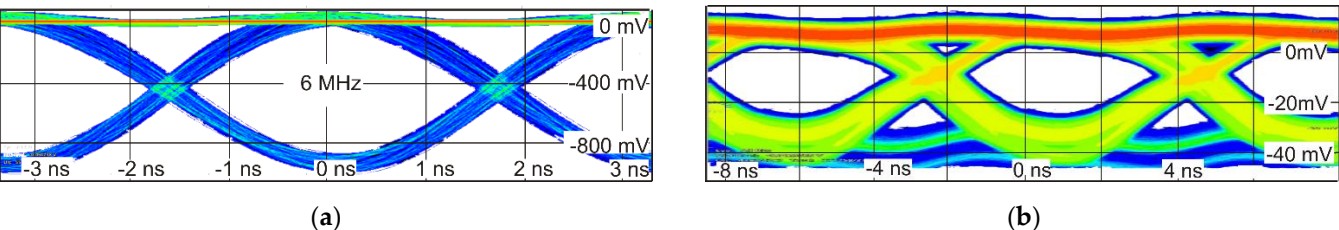

(**a**)                                                                                        (**b**)

**Figure 13.** View of the eye diagrams for FSO setup with (**a**) pulsed and (**b**) cw lasers.

## 4. Conclusions

In this study, the capabilities of state-of-the-art QC lasers for free-space communications operated in an MWIR spectral range are defined. The performed simulations show that weather conditions have a significant impact on data link performances. For "clear" atmosphere, a better link range can be obtained with 1.5 μm wavelength (SWIR) because of lower attenuation by water vapor. If visibility decreases and air turbulence increases, the longer range is provided by longer wavelengths (MWIR). According to the parameters of the designed FSO laboratory models with pulsed QC laser (~3 W peak power, ~4.5 μm wavelength) and with cw QC laser (~20 mW average power, ~4.8 μm wavelength), link budgets were compared. A longer link distance was determined for the pulsed laser resulting from its 150-times higher radiation power. However, this difference is negligible in the case of strong scintillations. A preliminary test of these lasers indicated that the pulse duty cycle was more critical for a pulsed laser. It was also noticed that the cw laser could be operated with a much higher peak power (10 times), but with lower duty cycle, and therefore a compromise between data rate and link distance of the FSO system should be confirmed. Finally, the eye diagram tests of the FSO systems with two lasers were performed. The results indicated that both QC lasers can be used in FSO systems providing a modulation rate of 6 MHz (for the pulsed laser) and 100 MHz (for the cw laser).

**Funding:** This research received no external funding.

**Institutional Review Board Statement:** Not applicable.

**Informed Consent Statement:** Not applicable.

**Data Availability Statement:** Not applicable.

**Acknowledgments:** I would like to thank my colleagues Dariusz Szabra and Zbigniew Zawadzki for their support. The research was carried out in the laboratory of the Institute of Optoelectronics MUT and supported by the Narodowe Centrum Badań i Rozwoju, grant no. MAZOWSZE/0196/19-00. The assistance of the Sieć Badawcza Łukasiewicz—Instytut Technologii Elektronowej (Kamil Pierściński) by providing support with the quantum cascade laser technology is gratefully acknowledged.

**Conflicts of Interest:** The author declares no conflict of interest.

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
