# Peer review of "A Comparison Study of Data Link with Medium-Wavelength Infrared Pulsed and CW Quantum Cascade Lasers"

_photonics, doi:10.3390/photonics8060203_

Round 1

Reviewer 1 Report

In this article, the authors make a comparative study between two types of QC lasers for free space optical communications. Both QC lasers are evaluated in terms of modulation rate, link range and peak power.  The results are useful as  reference to select laser source for free space optical communications. However, the article should be aimed to contribute some new designs or new findings, which are more expected from reader's viewpoint, for example, to discuss some potential opportunities for improving the lasers' performance in modulation rate and stability. Furthermore, it is suggested to make clear the figures for better readability. 

Author Response

At first, I would like to thank you for your suggestions and opinion. They are very accurate and significantly increase the research level of my work. Some responses about your comments are described in the cover letter and all the changes are highlighted (yellow) in the revised manuscript.  
Yours Faithfully
Janusz Mikolajczyk

Reviewer 2 Report

see the "comments.pdF" file.

Author Response

(The authors gave the same response as above.)

Reviewer 3 Report

Using longer wavelength ranges (medium-wavelength infrared (MWIR) could provide several advantages for FSO systems in eye safety and the reduction of scattering effects. This manuscript presents simulation-based performance analysis about the impacts of weather conditions on the link budget of MWIR-FSO systems. Also, presented is a preliminary test of MWIR lasers. 

 There are some minor comments on the manuscript:

1) There is missing data for distances in Table 1. You may revise the table.   

2) What is the equation to plot Fig. 4? You may add that equation.

3) In Fig. 12, what is the FSO setup to get the eye diagram? You may add more description about the setup.

4) The quality of the figures in the current manuscript is not that good. You may improve them. 

Author Response

(The authors gave the same response as above.)
